# Generation and Characterization of Anti-Filovirus Nucleoprotein Monoclonal Antibodies

**DOI:** 10.3390/v11030259

**Published:** 2019-03-14

**Authors:** Md Niaz Rahim, Min Wang, Tong Wang, Shihua He, Bryan D. Griffin, Darwyn Kobasa, Ruifu Yang, Zongmin Du, Xiangguo Qiu

**Affiliations:** 1Special Pathogens Program, National Microbiology Laboratory, Public Health Agency of Canada, 1015 Arlington Street, Winnipeg, MB, R3E 3R2, Canada; mdniaz.rahim@canada.ca (M.N.R.); shihua.he@canada.ca (S.H.); bryan.griffin@canada.ca (B.D.G.); darwyn.kobasa@canada.ca (D.K.); 2Department of Medical Microbiology, University of Manitoba, 745 Bannatyne Avenue, Winnipeg, MB, R3E 0J9, Canada; 3State Key Laboratory of Pathogen and Biosecurity, Beijing Institute of Microbiology and Epidemiology, Beijing 100071, China; yrf007@sina.com (M.W.); wtwangtongqz@163.com (T.W.); ruifuyang@gmail.com (R.Y.)

**Keywords:** mAb (monoclonal antibody), Ebola (EBOV), Sudan (SUDV), Bundibugyo (BDBV), Marburg (MARV), Tai Forest (TAFV), Reston (RESTV) viruses

## Abstract

Filoviruses cause lethal hemorrhagic fever in humans. The filovirus nucleoprotein (NP) is expressed in high abundance in infected cells and is essential for virus replication. To generate anti-filovirus monoclonal antibodies (mAbs) against the NP, mice were immunized with peptides known as B-cell epitopes corresponding to different filovirus NPs, and hybridomas were screened using FLAG-tagged filovirus NP constructs. Numerous mAbs were identified, isotyped, and characterized. The anti-NP mAbs demonstrated different ranges of binding affinities to various filovirus NPs. Most of the clones specifically detected both recombinant and wild-type NPs from different filoviruses, including Ebola (EBOV), Sudan (SUDV), Bundibugyo (BDBV), Marburg (MARV), Tai Forest (TAFV), and Reston (RESTV) viruses in western blot analysis. The mAbs were also able to detect native NPs within the cytoplasm of infected cells by immunofluorescence confocal microscopy. Thus, this panel of mAbs represents an important set of tools that may be potentially useful for diagnosing filovirus infection, characterizing virus replication, and detecting NP–host protein interactions.

## 1. Introduction:

Ebola virus (EBOV), an enveloped, negative-sense RNA virus, is a member of the *Ebolavirus* genus and the *Filoviridae* family. It is the causative agent of a severe and highly lethal hemorrhagic fever that has resulted in numerous outbreaks throughout Africa, including an unprecedented outbreak in 2014–2016 that resulted in more than 28,000 cases and more than 11,000 deaths [1]. Other ebolaviruses, including Sudan (SUDV) and Bundibugyo (BDBV) viruses, as well as Marburg (MARV) virus, which belongs to the related *Marburgvirus* genus within the family *Filoviridae*, are also highly virulent in humans and have caused numerous outbreaks [2]. Conversely, although Reston virus (RESTV) and Tai Forest virus (TAFV) are also ebolaviruses, the former appears to be apathogenic in humans, while the latter has caused only a single reported human infection [3]. No Food and Drug Administration (FDA)-approved vaccine or antiviral treatment is available for treating filovirus disease; however, several candidate vaccines and therapeutics, including monoclonal antibody therapeutics, have shown efficacy in clinical trials and are currently being deployed in the 2018 Democratic Republic of Congo EBOV outbreak [4,5,6,7,8].

Filoviruses encode seven different structural proteins: the nucleoprotein (NP), virion protein (VP) 35, VP40, VP24, the glycoprotein (GP), VP30, and the RNA-dependent RNA polymerase (L). The NP, which binds directly to the viral genome, plays a critical role in the viral replication cycle as the main component of the viral nucleocapsid, along with VP35, VP24, VP30, and L [9]. In a previous study, Changula et al. generated mouse monoclonal antibodies by immunizing mice with EBOV virus-like particle (VLP) NPs and mapped the epitopes. They then used a combination of empirical analyses and in silico prediction to identify B-cell epitopes for different filovirus NPs and generate rabbit polyclonal antisera specific to different filovirus NPs [10]. In another study, Becquart et al. also determined potential B-cell epitopes for the EBOV NP by analyzing human survivors’ serum [11]. Given that the NP is a critical and highly abundant viral protein, we sought to produce a set of mouse monoclonal antibodies specific for multiple different filovirus NPs by immunizing mice with these previously identified NP epitopes [10], with the hope that they might serve as tools for developing novel diagnostic tests and dissecting filovirus molecular virology.

## 2. Methods

### 2.1. Cell Lines and Viruses

HEK293T cells were maintained in Dulbecco’s modified Eagle’s medium (DMEM) or Roswell Park Memorial Institute (RPMI) 1640 medium containing 10% fetal bovine serum and 2 mM L-glutamine at 37 °C in a 5% CO2 incubator. Vero E6 cells obtained from the American Type Culture Collection (ATCC) were grown in DMEM supplemented with 1% Penicillin-Streptomycin and 10% heat-inactivated fetal bovine serum (FBS). CV-1 cells were grown in minimum essential medium (MEM) supplemented with 1% Penicillin-Streptomycin and 5% heat-inactivated FBS. EBOV (EBOV/H.sapiens-tc/GIN/14/Makona- Gueckedou-C07), MARV (H.sapiens-tc/AGO/ 2005/Angola), BDBV (H.sapiens-tc/UGA/07/But-811250) TAFV (TAFV/H.sapiens-tc/CIV/1994/Pauleoula-CI), and RESTV (RESTV/M.fascicularis-tc/USA/1990/Philippines89-AZ1435) were grown in Vero E6 cells in DMEM supplemented with 2% FBS and 1% Penicillin-Streptomycin in a biosafety containment level 4 (CL4) laboratory. SUDV (SUDV/H.sap-gp-tc/SDN/1976/Boneface-USAMRIID111808), variant Boneface, was grown in African green monkey kidney CV-1 cells in MEM supplemented with 1% Penicillin-Streptomycin and 2% heat-inactivated FBS in a CL4 laboratory. Viruses in the media supernatants were concentrated by ultracentrifugation at 64,000× *g* for 2 h at 4 °C, and viral pellets were resuspended with phosphate-buffered saline (PBS). The concentrated virus was then inactivated by gamma irradiation (7 Megarad) using the MDS Nordion Gammacell Excel 220 Cobalt-60 (^60^Co) source. All the experiments using live virus were performed at the Canadian Science Centre for Human and Animal Health (CSCHAH) containment level 4 laboratory by trained personal following approved procedures.

### 2.2. Production of Filovirus Antigens

A total of 6 filovirus NP peptide fragments (14–22 amino acids long; Table 1), which were determined as B-cell epitopes by Changula et al. [10] and Becquart et al. [11], were synthesized by Guoping Pharmaceutical Co. (Anhui, China). The purity of the peptides was >95%. Keyhole Limpet Hemocyanin (KLH)-conjugated peptides were used as antigens for the immunization of mice, and BSA-conjugated peptides were used in ELISA analysis of the secreted antibodies by hybridoma. FLAG-tagged, full-length EBOV, SUDV, RESTV, and TAFV NP expression constructs, under the control of a Cytomegalovirus (CMV) promoter, were synthesized according to the procedure described in Tan Y et al. [12]. HEK293T cells were transfected with NP-expressing pFLAG-CMV plasmids using Mega Tran 1.0 to express EBOV, SUDV, TAFV, and RESTV NPs, according to the manufacturer’s instructions.

### 2.3. Mice Immunization and Monoclonal Antibody (mAb) Production

All animal procedures and husbandry were conducted according to the Guidelines for the Welfare and Ethics of Laboratory Animals of China. Mice immunization and mAb production were conducted following the procedure described in Wang et al. [13]. In brief, 6–8 week-old female BALB/c mice were subcutaneously immunized with different filovirus NP peptides (Table 1). BALB/c mice (18–20 g) were immunized with 20 μg of individual peptides with complete Freund’s adjuvant. Then, 2 weeks later, the mice were injected with 50 μg of individual peptides with incomplete Freund’s adjuvant, and the immunization procedure was repeated once after 2 weeks with 50 µg of individual peptides. The spleen cells were removed 3 days after the boost and fused with SP2/0 myeloma cells. Hybridomas were cloned by limit dilution and screened using different synthesized peptides by ELISA. mAbs were isotyped using an IsoStrip^TM^ Mouse Monoclonal Antibody isotyping kit (Sigma, St Louis, MO, USA) and were purified by caprylic acid–ammonium sulfate precipitation of ascites [14]. Purified mAbs were evaluated via western blots analysis where FLAG-NP proteins were used.

### 2.4. Indirect ELISA

ELISAs were conducted in flat-bottom, 96-well plates (Corning, Corning, NY, USA). BSA-conjugated peptides were used in ELISA to screen hybridomas. Purified mAbs were tested against filovirus NPs by indirect ELISA. Each well was coated overnight at 4 °C with 40 ng of a different irradiated filovirus preparation, washed 3 times with PBS (pH 7.3), and blocked with 5% skim milk (Difco, BD Maryland, USA) for 1.5 h. Different concentrations of purified mAbs (3 μg–0.003 μg) in 5% skim milk were added to the well and incubated for 2 h at room temperature. The wells were washed 4 times with PBS supplemented with 0.1% tween 20 (PBST), and the bound antibodies were detected following incubation with horseradish peroxidase (HRP)-linked secondary goat anti-mouse IgG (Southern Biotech, Cat 1030-05) for 1 h and 3,3^′^,5,5^′^-tetramethylebenzidine (Life technology, Carlsbad, CA, USA) for 30 min at room temperature. The optical density (OD) was determined at 650 nm using an ELISA V_max_ kinetic microtiter plate reader. A PBS-only control was maintained on each plate.

### 2.5. Western Blot Analysis

The irradiated virus (10 μg) was mixed with LDS electrophoresis sample buffer (Life technology) and 100 μM Dithiothreitol (DTT), boiled for 7 min, and resolved in a 4–12% gradient NuPAGE SDS-PAGE Gel system (Life technology). Proteins were transferred to nitrocellulose membranes using the iBlot® 2 dry blotting system (Life technology), according to the manufacturer’s protocol. The membranes were blocked with Odyssey Blocking buffer (LI-COR) before being treated with purified anti-filovirus NP mAbs (1 mg/mL) diluted into 1:1000 in blocking buffer. The blots were probed with fluorescent-labeled goat anti-mouse monoclonal secondary antibody (IRDye 8 CW, LI-COR), and the signal was detected using an Odyssey (LI-COR) scanning system. FLAG-tagged filovirus NP proteins were also subjected to western blot analysis with an anti-FLAG antibody (Sigma-Aldrich, St Louis, MO, USA) to validate the expression.

### 2.6. Confocal Microscopy (CM)

Vero E6 and CV-1 cells were grown in 96-well plates, infected with different filoviruses at a multiplicity of infection (MOI) of 3, and fixed with 10% formaldehyde 48 h post-infection (hpi). MOI 3 was used to confirm that the majority of the cells were infected. Fixed cells were washed with PBS, permeabilized with 0.1% Triton X-100 in PBS for 5 min, and blocked with 3% BSA/PBS for 30 min. The cells were then incubated with different anti-filovirus NP mAbs (1 mg/mL) diluted to 1:300 in 1% BSA/PBS for 1.5 h at room temperature, followed by incubation with the secondary antibody Alexa Fluor 488-conjugated goat anti-mouse (Thermo fisher, Waltham, OH, USA) for 1 h. Nuclei were stained with DAPI (Thermo fisher). Images were acquired using a Zeiss LSM 700 confocal microscope (Jena, Germany).

## 3. Results

### 3.1. Anti-NP mAb Production and Isotyping

In order to produce a panel of NP-specific antibodies, we synthesized a set of NP peptides that were previously characterized as immunogenic (Table 1) [10]. The regions covered by EBOV NP aa421–440 displayed considerable similarity among all ebolaviruses and were used to generate pan-ebolavirus antibodies. The remaining peptides, EBOV NP aa491–510, SUDV NP aa631–644, TAFV NP aa630–643, RESTV NP aa630-643, and MARV NP aa635–652 corresponded to regions that showed little similarity among different filoviruses and were used to generate virus-specific antibodies [10]. Female BALB/c mice were immunized with the NP peptides (Table 1), after which spleen cells were collected and fused with myeloma cells, and potential monoclonal hybridomas were detected by ELISA using BSA-conjugated peptides. Positive hybridomas were cloned by sequential limiting dilution method, and supernatants were screened using western blot analyses against different recombinant FLAG-tagged NP proteins. In total, 20 different mAbs were identified and subsequently purified by protein G affinity chromatography (Table 2). Immunoglobulin isotyping showed that three mAbs were IgM and 17 were IgG (Table 2). Subtyping of the IgG antibodies revealed that two were IgG2b and 15 were IgG1 (Table 2).

### 3.2. Characterization of Anti-NP mAb Reactivity

To assess the efficacy of our purified anti-NP mAbs, a wide range of antibody concentrations were first tested against inactivated whole virus preparations of various filoviruses by ELISA (Figure 1). Most of the mAbs demonstrated binding capacity with different affinities to various filovirus NPs (Figure 1 and summarized in Table 2). The EBOV-specific anti-NP mAbs (15D10 and 6E5), as well as the SUDV-specific mAbs (3F12, 11G9, 6G5, and 12F12) demonstrated medium to strong binding to the NP. Both RESTV-specific mAbs (2A10 and 7E7) and all the TAFV-specific mAbs (4B10, 2A10, 8G9, and 1H3) were able to bind moderately to the NP. Of the MARV-specific mAbs, only 6E9, 8H4, and 7A12 showed strong binding affinity to the NP, whereas 1G10 and 5E9 bound weakly. Notably, preliminary screening revealed two antibodies (2G10 and 4F9) that were cross-reactive against multiple filovirus NPs. In our ELISA assay, antibody 2G10 bound strongly with the EBOV NP and moderately with the NP of SUDV, TAFV, and BDBV; however, it did not bind with the MARV NP (Figure 1B). Similarly, 4F9 bound the NP from all the filoviruses, except MARV, with varying affinities (Figure 1B).

We next sought to determine if the NP mAbs were able to detect denatured NP in a western blot assay. Inactivated whole virus preparations (10 μg) were resolved by SDS-PAGE and immunoblotted with the different anti-NP mAbs (Figure 2). Most of the mAbs detected filovirus NPs with a range of apparent affinities, reflecting what was observed in the ELISA assay (summarized in Table 2). EBOV-specific mAbs (15D10 and 6E5) demonstrated medium to strong reactivity to denatured EBOV NP (Figure 2A). All (3F12, 11G9, and 6G5) but one (12F12) SUDV-specific and all the TAFV-specific (4B10, 2A10, 8G9, and 1H3) mAbs demonstrated strong affinity to NP (Figure 2B,D). Both RESTV-specific mAbs (2A10 and 7E7) detected denatured NP with medium reactivity (Figure 2C), and the all MARV-specific mAbs (3E10, 6E9, 5E9, 8H4, and 7A12) except one (1G10) reacted strongly with the MARV NP (Figure 2E). The reactivity of MARV-specific 6E9, 8H4, and 7A12 in our western blot assay was consistent with the ELISA results (Figure 1A and Figure 2E). Both of the cross-reactive mAbs (2G10 and 4F9) demonstrated strong reactivity to the EBOV and BDBV NPs, weak reactivity to the SUDV and TAFV NPs, and no reactivity to the RESTV and MARV NPs (Figure 2F,G). Notably, although the predicted molecular weight of the NP is ~85 kDa, all the mAbs detected a band corresponding to the NP around 100–120 kDa, consistent with a previous report [15]. Moreover, in some cases, additional low molecular weight bands appeared in our western blot that may represent proteolytic degradation and secondary site translation initiation.

Because gamma irradiation can disrupt viral structure and lead to the denaturation of viral proteins, we wanted to assess the reactivity of our antibodies against native antigens. To this end, we infected Vero E6 and CV-1 cells at a MOI of 3 with EBOV, SUDV, TAFV, BDBV, or MARV for 48 h and then used our panel of mAbs for CM (Figure 3). Most of the mAbs were able to detect the filovirus NP in its native form, with the results generally agreeing with the ELISA and western blots (Table 2). The NP was found to be distributed throughout the cytoplasm, which is consistent with previous findings [16] (Figure 3). Both EBOV NP-specific mAbs (15D10 and 6E5) detected native NP without any background in mock-infected cells (Figure 3A). Likewise, the SUDV-specific (3F12, 11G9, 6G5, and 12F12) and RESTV-specific (2A10 and 7E7) mAbs recognized the NP, albeit with fainter reactivity (Figure 3B,C). One of the TAFV-specific mAbs (1H3) recognized NP with exceptional affinity, while the other three (2A10, 4B10, and 8G9) worked poorly (Figure 3D), despite working well in the ELISA and western blot assays (Figure 1A and Figure 2D). The majority of the MARV-specific mAbs (3E10, 6E9, 5E9, 8H4, and 7A12) recognized the native MARV NP with medium to strong reactivity except 1G10 and 3E10 (Figure 2E). Although 3E10 recognized the denatured MARV NP with high reactivity in the ELISA and western blot assays (Figure 1A and Figure 2E), it showed very weak affinity for the native MARV NP in immunofluorescence (Figure 3E; Table 2). Conversely, antibody 1G10 demonstrated the weakest reactivity in the ELISA and western blot assays (Figure 1A and Figure 2C) and very little signal in immunofluorescence in CM (Figure 3C), indicating a weak binding capacity for the MARV NP. The cross-reactive antibody 2G10 recognized all the filovirus NPs, except that of MARV (Figure 3F), whereas 4F9 was able to detect the native NP from EBOV, SUDV, and RESTV, as well as MARV, although the immunofluorescence signal in the latter case was weak (Figure 3F). Interestingly, both of the cross-reactive mAbs detected the native RESTV NP by immunofluorescence (Figure 3F) but were unable to detect the denatured NP in western blot and only weakly detected the NP in the ELISA assay (Figure 1 and Figure 2).

## 4. Discussion

The monoclonal antibodies against the filovirus NPs that were produced and characterized herein represent an important set of tools that may be used to develop novel diagnostic assays or to dissect the molecular biology of these viruses. Together, our panel of 20 antibodies recognized the NPs from at least one virus belonging to each filovirus species (Table 2), with the exception of Lloviu and Bombali viruses, which were not tested. Most of the antibodies functioned well in multiple assays, demonstrating reactivity against both linear NP antigens (western blot) and conformationally native NPs (confocal microscopy). Moreover, in another study, we demonstrated the ability of our anti-RESTV NP mAb clone 7E7 to detect RESTV by immunohistochemistry in a ferret model of infection [17]. As expected, the reactivities of our mAbs targeting specific filovirus NP peptide sequences correlated with those in the Changula et al. study [10]; however, we were able to generate multiple mAb clones for each filovirus NP.

Although the majority of antibodies were specific for the NP of a specific filovirus, we did identify and characterize two antibodies, 2G10 and 4F9, that displayed cross-reactivity among multiple ebolavirus NPs. Antibody 2G10, worked particularly well in the immunofluorescence assay, recognizing EBOV, RESTV, TAFV, and, to a lesser extent, SUDV. Interestingly, this antibody was unable to recognize the RESTV NP in the western blot assay, and it only weakly bound to the RESTV NP in the ELISA. Conversely, 2G10 was highly reactive to the BDBV NP in the ELISA and western blot assays; however, BDBV was not tested against 2G10 by immunofluorescence assay. Antibody 4F9 displayed similar reactivity patterns to 2G10, recognizing EBOV, SUDV, and BDBV NP the best by ELISA and western blot assays but failing to bind to the RESTV NP by western blot and only poorly binding to the TAFV NP. Curiously, both of these antibodies detected multiple bands (ranging in size from ~110 kDa to ~70 kDa) for the SUDV NP in the western blot assay. This pattern of reactivity is distinct from that observed with the SUDV-specific antibodies (3F12, 11G9, 6G5, and 12F12), which each detected a major band around ~110 kDa, and the cause is unclear. We suspect that differences in antibody epitopes may result in differences in recognition of NP degradation products or isoforms, but we note that 2G10 and 4F9 still recognize the ~110 kDa isoform of the NP. Importantly, our characterization of all 20 antibodies described in this study used a single, standardized protocol for each assay, and further optimization of the experimental conditions for each individual antibody will be required to improve their efficacy, especially as it relates to their diagnostic utility.

Filoviruses, particularly EBOV, SUDV, BDBV, and MARV, represent significant global public health threats, yet several gaps still exist in our understanding of these viruses and our ability to rapidly diagnose and treat the infections they cause. Rapid and reliable diagnosis of filovirus infections plays a crucial role in limiting the spread of disease during an outbreak [18]. Given that the NP is among the most highly expressed filoviral proteins [19], it represents an attractive target for developing novel serological diagnostic tools. Indeed, future work will focus on developing a potentially pan-filovirus diagnostic test by selecting and pooling the most effective mAbs described here. Ideally, such a diagnostic test would be easily and inexpensively deployed in the field and able to distinguish among multiple different filoviruses. In addition to being potentially useful for virus diagnosis, these antibodies are likely to also prove invaluable in dissecting the molecular biology of filoviruses. The NP is a major structural protein that plays a vital role in virus replication [20,21,22], yet relatively little else is known about other roles this protein may have or, in particular, the host proteins with which it interacts.

## Figures and Tables

**Figure 1 viruses-11-00259-f001:**
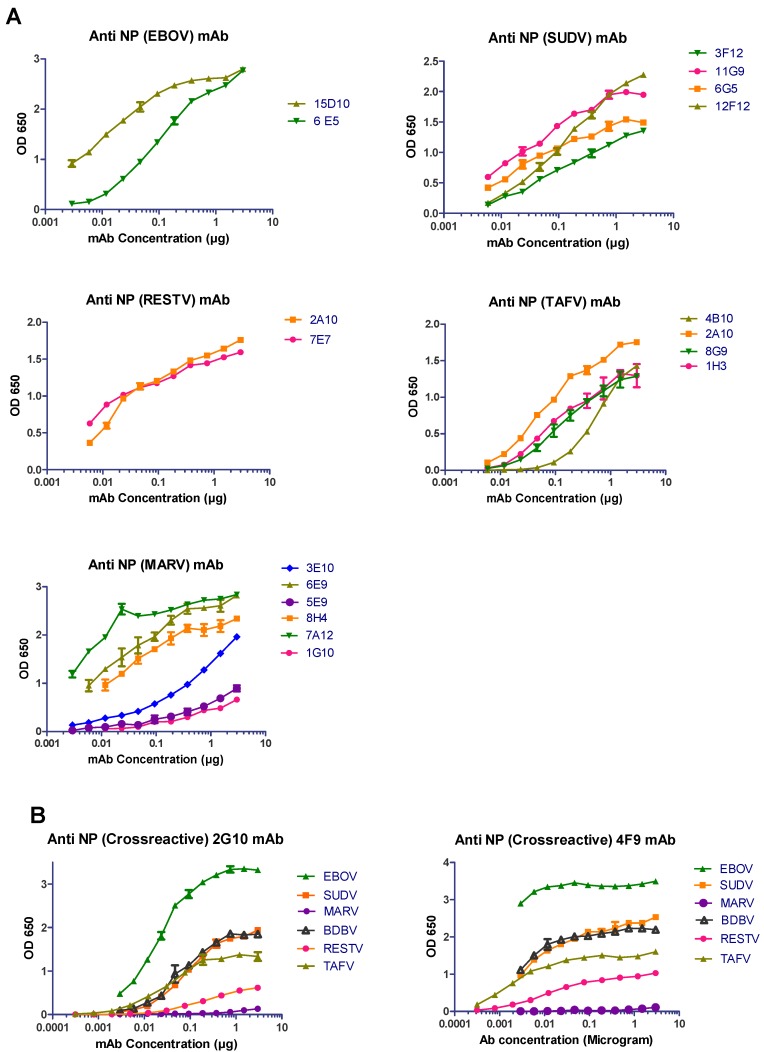
Anti-NP mAbs bind with wild-type filovirus NPs. (**A**) ELISA plates were coated with 40 ng of purified irradiated filoviruses EBOV, SUDV, RESTV, TAFV, and MARV. Different concentrations of purified mAbs (3 μg–0.003 μg per well) were added in respective plates. After blocking, the bound antibodies were detected following incubation with horseradish peroxidase (HRP)-linked secondary goat anti-mouse IgG. The optical density (OD) was determined at 650 nm. (**B**) ELISA plates coated with 40 ng of EBOV, SUDV, MARV, BDBV, RESTV, and TAFV were treated with different concentrations of purified 2G10 and 4F9 mAbs (3 μg–0.003 μg per well) and analyzed via binding assay.

**Figure 2 viruses-11-00259-f002:**
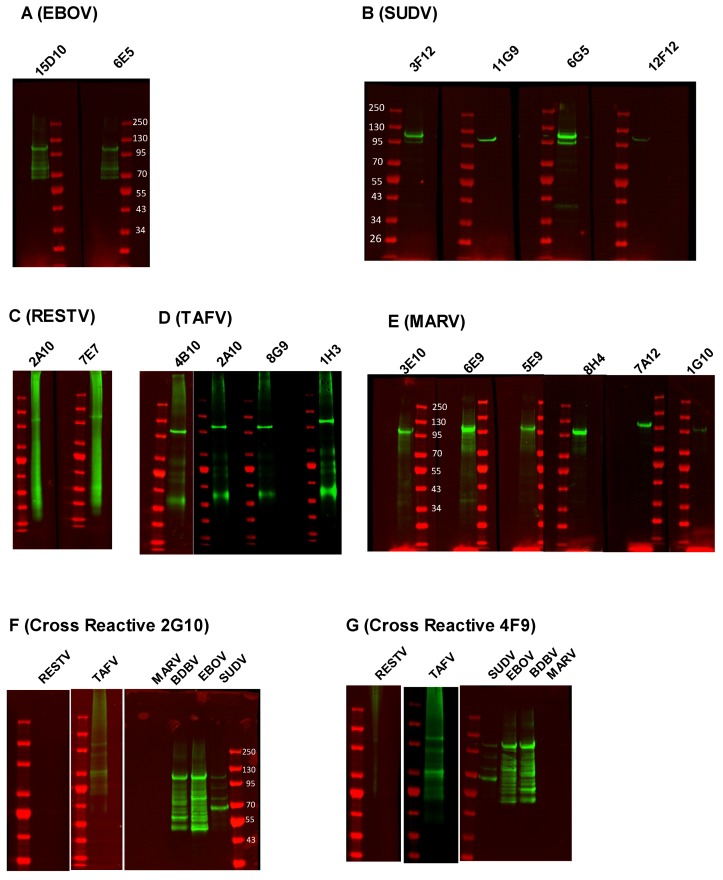
Western blot analysis of anti-filovirus mAbs against wild-type filovirus NP. 10 µg of purified irradiated EBOV (**A**), SUDV (**B**), RESTV (**C**), TAFV (**D**), and MARV (**E**) were resolved in SDS-PAGE gels, and the proteins were transferred into nitrocellulose membranes. The membranes were treated with (A) 15D10 and 6E5; (B) 3F12, 11G9, 6G5, and 12F12; (C) 2A10 and 7E7; (D) 4B10, 2A10, 8G9, and 1H3; (E) 3E10, 6E9, 5E9, 8H4, 7A12, and 1G10; mAbs. Fluorescent-labeled goat anti-mouse monoclonal secondary antibodies detected the NP. 10 µg of purified RESTV, TAFV, SUDV, EBOV, BDBV, and MARV were resolved into two separate SDS-PAGE gels, the resolved proteins were transferred into nitrocellulose membrane, and the blots were treated with 2G10 (**F**) and 4F9 (**G**) mAbs to detect different filovirus NPs.

**Figure 3 viruses-11-00259-f003:**
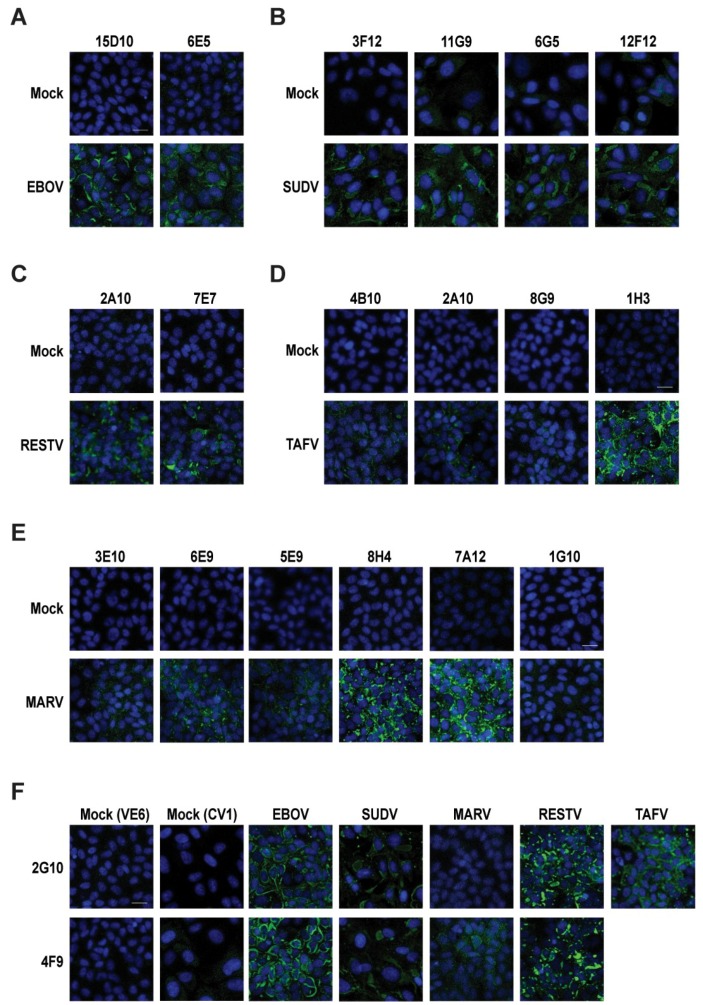
Anti-NP filovirus mAbs detected the native filovirus NP in confocal microscopy (CM). In CM, the cells were infected with a multiplicity of infection (MOI) of 3, fixed and permeabilized at 48 h post-infection (hpi), and stained with different anti-filovirus NP mAbs; the nuclei were stained with DAPI, and finally filovirus NP expressions were detected by treating with Alexa 488 secondary anti-mouse antibodies (Abs). Mock infection controls were also maintained with both primary and secondary Ab staining. (**A**) Vero E6 cells were infected with EBOV treated with 15D10 and 6E5 mAbs. (**B**) CV1 cells were infected with SUDV and treated with 3F12, 11G9, 6G5, and 12F12 mAbs. (**C**) Vero E6 cells were infected with RESTV and stained with primary 2A10 and 7E7 mAbs. (**D**) Vero E6 cells were infected with TAFV and treated with 4B10, 2A10, 8G9, and 1H3 mAbs. (E) Vero E6 were infected with MARV and treated with 3E10, 6E9, 5E9, 8H4, 7A12, and 1G10 mAbs. (**F**) CV1 cells were infected with SUDV, and Vero E6 cells were infected with EBOV, MARV, RESTV, and TAFV; therefore, they were treated separately with 2G10 and 4F9 to check cross reactivities against different filoviruses. The scale bar represents 20 µm.

**Table 1 viruses-11-00259-t001:** Filovirus nucleoprotein (NP) peptides sequences for mice immunization.

Target	Amino Acid Sequences	mAbs Generated
EBOV-NP	aa 421–440: YDDDDDIPFPGPINDDDNPG	4F9, 2G10
EBOV-NP	aa 491–510: DDEDTKPVPNRSTKGGQQKN	6E5, 15D10
SUDV-NP	aa 631–644: QGSESEALPINSKK	6G5, 3F12, 11G9, 12F12
TAFV-NP	aa 630–643: NQVSGSENTDNKPH	1H3, 8G9, 4B10, 2A10
RESTV-NP	aa 630–643: TSQLNEDPDIGQSK	7E7, 2A10
MARV-NP	aa 635–652: RVVTKKGRTFLYPNDLLQ	1G10, 3E10, 5E9, 6E9, 8H4, 7A12

**Table 2 viruses-11-00259-t002:** Characteristics of anti-filovirus NP mAbs.

mAb	Type	Reactivity	ELISA (Irradiated Virus)	Western Blot (Irradiated Virus)	Confocal Microscopy (Wild-Type Infection)
15D10	IgG1	EBOV	++	++	++
6E5	IgG1		++	++	++
3F12	IgG1	SUDV	++	+++	+
11G9	IgG1		+++	+++	++
6G5	IgG1		++	+++	++
12F12	IgG1		+++	+	++
2A10	IgG1	RESTV	++	++	++
7E7	IgG1		++	++	++
4B10	IgG2b	TAFV	++	+++	+
2A10	IgG1		++	+++	+
8G9	IgG1		++	+++	−
1H3	IgG1		++	+++	+++
3E10	IgM	MARV	++	+++	+/−
6E9	IgG2b		+++	+++	++
5E9	IgM		+	+++	+
8H4	IgG1		+++	+++	+++
7A12	IgG1		+++	+++	+++
1G10	IgG1		+	+	−
2G10	IgG1	EBOV	+++	+++	+++
		SUDV	++	+	++
		TAFV	++	+	++
		RESTV	+	−	+++
		MARV	−	−	−
		BDBV	++	+++	ND
4F9	IgM	EBOV	+++	+++	+++
		SUDV	+++	+	++
		TAFV	++	+	ND
		RESTV	+	−	+++
		MARV	−	−	+
		BDBV	++	+++	ND

+++ Highly reactive, ++ Moderately Reactive, + Weakly reactive, − non-reactive, ND not done.

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
