# Peer review of "Generation and Characterization of Anti-Filovirus Nucleoprotein Monoclonal Antibodies"

_viruses, 2019, doi:10.3390/v11030259_

Reviewer 1 Report

The manuscript titled, “Generation and Characterization of anti-Filovirus NP Monoclonal Antibodies” (Manuscript ID: viruses-448579) describes the generation and characterization of anti-filovirus monoclonal antibodies against nucleoprotein. The manuscript is well written and the methodology/data is scientifically sound. This newly generated panel of anti-filovirus monoclonal antibodies will be useful for application in future scientific studies. Listed below are a few minor comments.

Minor comments

Line 44: In order to make this manuscript easy to follow for years to come, please provide a few identifying details about the current outbreak in the Democratic Republic of the Congo (i.e., etiological agent, year outbreak began).

Line 49: Please include a bit more information regarding the empirical and in silico prediction analyses used by Changula et al. to identify B-cell epitopes for different filovirus NPs.

 Line 79 and Table 1: Only six filovirus NP peptide fragments are listed in Table 1, not eight as the manuscript text indicates. In addition, the target for the first peptide shown on page 3 is not indicated.

Line 96: Were the mice immunized a total of three times? If so, what dose of individual peptides was administered on the third inoculation?

Lines 137 and 139: EBOV NP aa601-620 and EBOV NP aa611-630 peptides are not included in Table 1.

Throughout the results and figures: To allow readers to easily follow the text, it would be helpful if the monoclonal antibodies for each filovirus were listed in the same order in the text and figures. For example, in the text TAFV-specific antibodies are listed in the following order: 1H3, 2A10, 4B10, and 8G9. While in Figure 2, the TAFV-specific antibodies are listed in the following order: 4B10, 2A10, 8G9 and 1H3.

Figure 2: The following part of the legend does not seem to match the diagram: “Membranes were treated with…”.

Author Response

REVIEWER 1:

We thank reviewer #1 for the time spent reviewing this manuscript.

Minor comments

Line 44: In order to make this manuscript easy to follow for years to come, please provide a few identifying details about the current outbreak in the Democratic Republic of the Congo (i.e., etiological agent, year outbreak began).

Response: As suggested, we have edited the 44 and mentioned the year, country and etiological agent (filovirus) of the outbreak.

Line 49: Please include a bit more information regarding the empirical and in silico prediction analyses used by Changula et al. to identify B-cell epitopes for different filovirus NPs.

            Response: As suggested, we have described in brief what Changula et al. conducted in their study in lines 49-54.

 Line 79 and Table 1: Only six filovirus NP peptide fragments are listed in Table 1, not eight as the manuscript text indicates. In addition, the target for the first peptide shown on page 3 is not indicated.

            Response: We apologize for the error. We used 6 filovirus NP fragments. Line 82 has been corrected. The target of the peptide aa491-510 has been mentioned in table 1 of page 3.

Line 96: Were the mice immunized a total of three times? If so, what dose of individual peptides was administered on the third inoculation?

            Response: Yes, the mice were immunized a total of three times. The third dose was the same as the second dose, which was 50 µg. We have included the dose in line 100.

Lines 137 and 139: EBOV NP aa601-620 and EBOV NP aa611-630 peptides are not included in Table 1.

Response: Again we apologize for the error. We used six filovirus peptides, as indicated in table 1. EBOV NP 601-620 and EBOV NP 611-630 were not used in this study. We have edited lines 141-144.

Throughout the results and figures: To allow readers to easily follow the text, it would be helpful if the monoclonal antibodies for each filovirus were listed in the same order in the text and figures. For example, in the text TAFV-specific antibodies are listed in the following order: 1H3, 2A10, 4B10, and 8G9. While in Figure 2, the TAFV-specific antibodies are listed in the following order: 4B10, 2A10, 8G9 and 1H3.

            Response:  We thank reviewer 1 for the comment. We have organized the names of filoviruses and monoclonal antibodies in Figures 1, 2 and Table 2 in the same order. The names of mAbs have been mentioned in similar order in the result sections of figure 1 and 2 (Lines 176-178, 199 to 202).

Figure 2: The following part of the legend does not seem to match the diagram: “Membranes were treated with…”.

Response: We have made corrections in the legend of Figure 2.

Reviewer 2 Report

This manuscript describes a set of experiments focused on the generation and characterization of anti-Filovirus NHP monoclonal antibodies.  The authors indicate a primarily diagnostic application of the technology/data described.  While of interest in the filovirus field, there are some flaws that need to be addressed.  Please see the following:

Throughout the document, there are grammatical errors (e.g., punctuation, sentence structure, proper use of grammar, etc.) that need to be revised.  A thorough review is warranted.

Tables 1 and 2 are broken between two pages; please ensure that tables are not split by pages

Table 1:  were the NP peptides verified as pure following production?

Line 75:  Please specify the gamma irradiation parameters.

Line 126:  The MOI of 3 is very high.  It is assumed that this is because of the short duration of infection (48h).  This requires further explanation.

Line 125:  Please explain the reasoning behind use of CV-1 cells.  All filoviruses (including SUDV) propagate well in Vero E6 cells.

Table 2:  Please explain how the antibody subclass (Type) was determined.  In addition, the specified reactivity classes (high, moderately, and weakly) are subjective; please  define further.

Figure 2:  The figure legend for Figure 2 does not align with the figure.  For instance, Figure 2A indicates EBOV with mAb 15D10 and 6E5, but the legend specifies 15D10, 6E5, 3F10, and 14E2.  Please inspect figures carefully (including Figure 3).

General question:  Why was BDBV not tested against 2G10 via immunofluorescence assay?  It would be helpful to see these data as well.

Lines 265-266:  The distinct pattern of reactivity described here warrants further explanation/discussion.  Although the authors note that the cause is unclear, there needs to be some expansion on this conclusion.  

Line 268:  Further optimization is needed before these mAbs can be used in any diagnostic sense; please strengthen this comment and remove "we suspect".

Author Response

Comments and Suggestions for Authors

This manuscript describes a set of experiments focused on the generation and characterization of anti-Filovirus NHP monoclonal antibodies.  The authors indicate a primarily diagnostic application of the technology/data described.  While of interest in the filovirus field, there are some flaws that need to be addressed.  Please see the following:

Response: We thank reviewer #2 for the time spent reviewing this manuscript.

Throughout the document, there are grammatical errors (e.g., punctuation, sentence structure, proper use of grammar, etc.) that need to be revised.  A thorough review is warranted.

Response: The manuscript has been checked by a native English speaking Canadian Scientist.

Tables 1 and 2 are broken between two pages; please ensure that tables are not split by pages

Response: Tables 1 and 2 have been reconstructed.

Table 1:  were the NP peptides verified as pure following production?

Response: Yes, the peptides were synthesized and purified by HPLC by the Guoping Pharmaceutical Co. (Anhui, China). The company tested the purity and certified that each peptide was >95% pure. We have indicated this in lines 84-85   

Line 75:  Please specify the gamma irradiation parameters.

Response: The gamma irradiation parameters are mentioned in lines 77-78.

Line 126:  The MOI of 3 is very high.  It is assumed that this is because of the short duration of infection (48h).  This requires further explanation.

Response: Yes, our target was to infect most of the cells so that we could stain and observe enough filovirus protein expression within 48 hours. Line 131-132 were updated.

Line 125:  Please explain the reasoning behind use of CV-1 cells.  All filoviruses (including SUDV) propagate well in Vero E6 cells.

Response: In our hands SUDV grows best in CV-1 cells and we have been using this CV-1 cells for growing SUDV stocks for many years. That was why we used CV-1 cells for SUDV.

Table 2:  Please explain how the antibody subclass (Type) was determined.  In addition, the specified reactivity classes (high, moderately, and weakly) are subjective; please define further.

Response: IsoStrip™ Mouse Monoclonal Antibody Isotyping Kit was used for antibody subclass typing and it is mentioned in Methodology section 2.3.

In this project we generated a panel of anti-filovirus NP mAbs and compared the reactivities among all mAbs. That was why we infected all cell lines with the same MOI, used the same amount of viral protein for western blotting and ELISA, and used the same same concentration of mAbs. Therefore, the strongest signals in western/confocal/ELISA were indicated as “high”. On the other hand, when the same number/amount of viruses/viral protein demonstrated very weak signals, we denoted this as weak reactivity.

Figure 2:  The figure legend for Figure 2 does not align with the figure.  For instance, Figure 2A indicates EBOV with mAb 15D10 and 6E5, but the legend specifies 15D10, 6E5, 3F10, and 14E2.  Please inspect figures carefully (including Figure 3)

Response: We apologize for the errors. The legends of Figure 2 and 3 have been edited.

General question:  Why was BDBV not tested against 2G10 via immunofluorescence assay?  It would be helpful to see these data as well.

Response: Yes, we agree. At that point we were in the process of purifying more mAbs including 2G10 and we also needed to grow more BDBV, that was why we could not include the 2G10 experiment with BDBV.

Lines 265-266:  The distinct pattern of reactivity described here warrants further explanation/discussion.  Although the authors note that the cause is unclear, there needs to be some expansion on this conclusion.

Response: Thanks for your response. We have added further explanations in lines 284-287  

Line 268:  Further optimization is needed before these mAbs can be used in any diagnostic sense; please strengthen this comment and remove "we suspect".

Response: We have removed the “we suspect” and re-written the sentence in lines 288-291.

 Round  2

Reviewer 2 Report

No further comments.  The authors have adequately addressed my comments and suggestions.